# Extreme magnetoresistance at high-mobility oxide heterointerfaces with dynamic defect tunability

D. V. Christensen [1] ✉, T. S. Steegemans[1], T. D. Pomar [1], Y. Z. Chen [1,2], A. Smith [1], V. N. Strocov [3], B. Kalisky [4] & N. Pryds [1]

Magnetic field-induced changes in the electrical resistance of materials reveal insights into the fundamental properties governing their electronic and magnetic behavior. Various classes of magnetoresistance have been realized, including giant, colossal, and extraordinary magnetoresistance, each with distinct physical origins. In recent years, extreme magnetoresistance (XMR) has been observed in topological and non-topological materials displaying a non-saturating magnetoresistance reaching $10^3$–$10^8$% in magnetic fields up to 60 T. XMR is often intimately linked to a gapless band structure with steep bands and charge compensation. Here, we show that a linear XMR of 80,000% at 15 T and 2 K emerges at the high-mobility interface between the large band-gap oxides $\gamma$-$Al_2O_3$ and $SrTiO_3$. Despite the chemically and electronically very dissimilar environment, the temperature/field phase diagrams of $\gamma$-$Al_2O_3$/$SrTiO_3$ bear a striking resemblance to XMR semimetals. By comparing magnetotransport, microscopic current imaging, and momentum-resolved band structures, we conclude that the XMR in $\gamma$-$Al_2O_3$/$SrTiO_3$ is not strongly linked to the band structure, but arises from weak disorder enforcing a squeezed guiding center motion of electrons. We also present a dynamic XMR self-enhancement through an autonomous redistribution of quasi-mobile oxygen vacancies. Our findings shed new light on XMR and introduce tunability using dynamic defect engineering.

The magnetic field-induced change in the electrical resistivity of materials and electronic devices may arise from a plethora of physical phenomena that sheds light on the underlying material properties, ranging from its spin state[1,2] and band structure[3,4] to its topological character[4], and quantum transport behavior[5,6]. Magnetoresistance (MR) phenomena are grouped into different classes, including giant[7,8], colossal[2] and extraordinary[9,10] magnetoresistance. More recently, extreme magnetoresistance (XMR) has been realized in several non-magnetic materials exhibiting semimetallic or other gapless behavior[4].

XMR manifests itself as a positive magnetoresistance far exceeding 1000%. A key characteristic is the lack of saturation in the magnetoresistance at high magnetic fields with, e.g., the XMR in semimetallic $WP_2$ not showing any signs of saturation up to 60 T where the magnetoresistance reaches a value of $2 \cdot 10^8$%[11]. By analyzing the temperature and field dependence of the resistance in chemically dissimilar gapless materials, Tafti et al. argue that a characteristic triangular temperature/field phase diagram is observed universally in XMR materials[12].

[1]Department of Energy Conversion and Storage, Technical University of Denmark, DK-2800 Kongens Lyngby, Denmark. [2]Beijing National Laboratory for Condensed Matter Physics and Institute of Physics, Chinese Academy of Sciences, Beijing 100190, China. [3]Swiss Light Source, Paul Scherrer Institute, 5232 Villigen-PSI, Switzerland. [4]Department of Physics and Institute of Nanotechnology and Advanced Materials, Bar-Ilan University, Ramat-Gan 5290002, Israel. ✉e-mail: dechr@dtu.dk

A universal mechanism for XMR remains elusive, but it is commonly observed to be intimately linked to the band structure[4]. For materials with a gapless band structure, $MR = \mu_e\mu_h B^2$ close to the charge compensation region characterized by an equal number of electrons and holes[4]. Steep electronic bands further enhance the electron mobility ($\mu_e$) and hole mobility ($\mu_h$) and boost the magnetoresistance in most XMR materials[4]. For uncompensated materials where the band structure facilitates a low effective mass and high carrier mobility, a non-saturating magnetoresistance with a positive linear dependence on the magnetic field may be observed[13–17]. The linear field dependence is often attributed to scattering or inhomogeneities in the magnetoresistive materials in a quantum[5,6], classical[18] or semi-classical[19] transport regime.

In the extreme quantum limit characterized by $\hbar\omega_c \gg k_B T$ and $\hbar\omega_c \gg E_F$ where $\omega_c$ and $E_F$ denote the cyclotron frequency and Fermi energy, respectively, only the lowest-lying Landau level is occupied. In the presence of scattering centers this has been predicted[5,6] and experimentally verified[15,20,21] to produce a linear, non-saturating magnetoresistance. In the classical transport regime of a strongly disordered material, a random resistor network subjected to spatial fluctuations in the mobility was used to simulate the magnetoresistance of an inhomogeneous conductor[18,22]. Such classical random resistor network models have subsequently been used to describe the linear magnetoresistance across electronic systems of different dimensionality and material classes[15,18,23–26].

Linear magnetoresistance also arises in the semi-classical transport regime of a weakly disordered material where the disorder potential is small compared to the Fermi energy and varies slowly on the scale of the cyclotron radius[19]. For materials without strict 2D confinement such as 3D Dirac materials, the disorder potential can prevent the free motion parallel to the field ($z$-direction) and constrain the current to flow with a slow guiding center motion in the $x/y$ plane superimposed by fast cyclotron orbits[19]. The current flow is squeezed between the inhomogeneities, which causes a linear increase in the magnetoresistance.

The magnetoresistive properties of complex oxides have been thoroughly investigated, in particular in manganites displaying a negative colossal magnetoresistance[2]. During the last two decades, the interest has been revived with the emergence of SrTiO$_3$-based heterostructures where 2-dimensionally confined electron gases exhibit strongly correlated transport with gate-tunable superconductivity coexisting with ferromagnetism[27,28], electron pairing in absence of superconductivity[29] and linear, unsaturated magnetoresistance[23–25] with gate-tunable transitions from a negative to positive magnetoresistance[30]. The electron gases located at the interface of SrTiO$_3$-based oxide heterostructures possess a set of very different electronic characteristics than the archetypical XMR materials. An

important example is the high-mobility electron gas at the γ-Al$_2$O$_3$/SrTiO$_3$ heterointerface. Here, the electronic properties are determined by a large bandgap structure where a large density of exclusively $n$-type carriers occupies heavy bands[31], possesses strong electron correlations[32,33] and confines in proximity to the interface according to the dynamic potential landscape formed by mobile electron donors[34–36]. Exploration of XMR phenomena in oxide heterointerfaces may therefore not only offer an alternative view on the mechanisms underlying XMR but also provide novel tuning capabilities allowing for investigating the XMR response to changes in the effective dimensionality and disorder landscape. The observation of linear magnetoresistance in various oxide material systems[23–25,37,38] and the recent observation of XMR in gapless and ambipolar SrNbO$_3$ thin films[39] with a magnetoresistance of 150,000% at 14 T have marked an exciting step towards realizing a high, positive magnetoresistance in oxide systems, but an experimental realization of XMR in correlated, high-mobility oxide heterointerfaces remains elusive.

Here, we present a large, unsaturated XMR exceeding 80,000% at 15 T in the strongly correlated γ-Al$_2$O$_3$/SrTiO$_3$ heterostructure. We show that the XMR can be tuned using defect engineering via three different approaches and use this concept to vary the XMR dynamically by several orders of magnitude. Lastly, we combine magneto-transport, dynamic defect engineering, microscopic visualization of current distributions, and momentum-resolved band structures to assign the high, linear XMR to a semiclassical guiding center motion of electrons in an inhomogeneous conductor and mechanistically explain how defect engineering tunes the XMR.

## Extreme magnetoresistance at oxide heterointerfaces

The γ-Al$_2$O$_3$/SrTiO$_3$ heterostructures are formed by pulsed laser deposition of epitaxial γ-Al$_2$O$_3$ on TiO$_2$-terminated, single-crystalline SrTiO$_3$ (001)-oriented substrates (see Methods). Under the current fabrication conditions, the deposition of γ-Al$_2$O$_3$ creates oxygen vacancies which produce conductivity in the interface-near region of SrTiO$_3$[35,40–42]. The magnetoresistance of the heterostructure, defined using the sheet resistance ($R_s$) as $MR = (R_s(B) - R_s(0\,T))/R_s(0\,T)$, reaches a value exceeding 80,000% at 2.4 K in a magnetic field of 15 T applied out-of-plane (see Fig. 1a). This value, reached through careful growth optimization, exceeds those in previous reports on magnetoresistance in γ-Al$_2$O$_3$/SrTiO$_3$[25] and other SrTiO$_3$-based heterointerfaces[23,43,44]. Second only to ambipolar SrNbO$_3$ thin films[39], it is one of the highest positive magnetoresistances observed across the broad class of oxide materials. Fitting to a power law function reveals that the magnetoresistance at 2.4 K scales as $MR \propto |B|^{1.51 \pm 0.01}$ at low magnetic fields up to a characteristic crossover field ($B_c$) of around 3 T (see Supplementary Fig. S1a). At higher magnetic fields, the

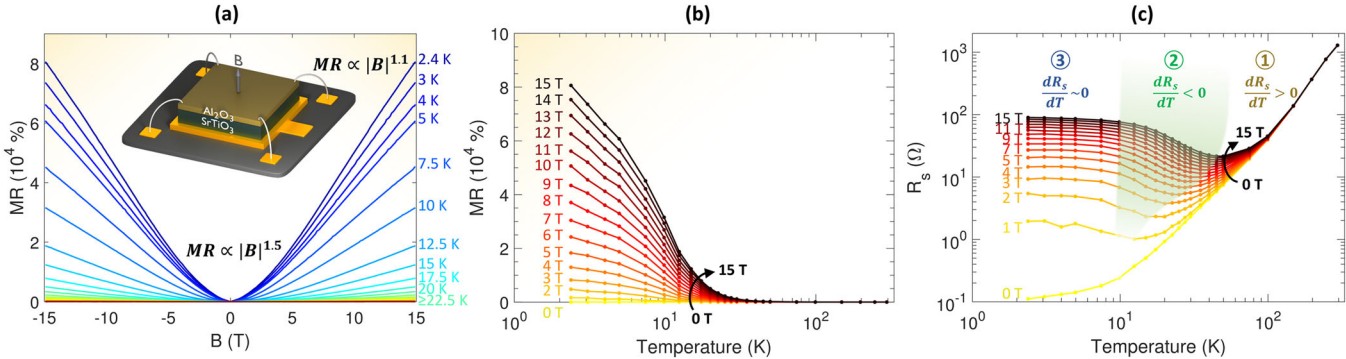

**Fig. 1 | Magnetoresistance in γ-Al$_2$O$_3$/SrTiO$_3$. a** Magnetoresistance ($MR$) of the γ-Al$_2$O$_3$/SrTiO$_3$ heterostructure displayed as a function of a magnetic field ($B$) applied normal to the interface. **b** Temperature dependence of the magnetoresistance showing a large increase at low temperatures and high magnetic fields. **c** Sheet resistance ($R_s$) as a function or temperature with three characteristic regions marking distinct magnetoresistive behavior.

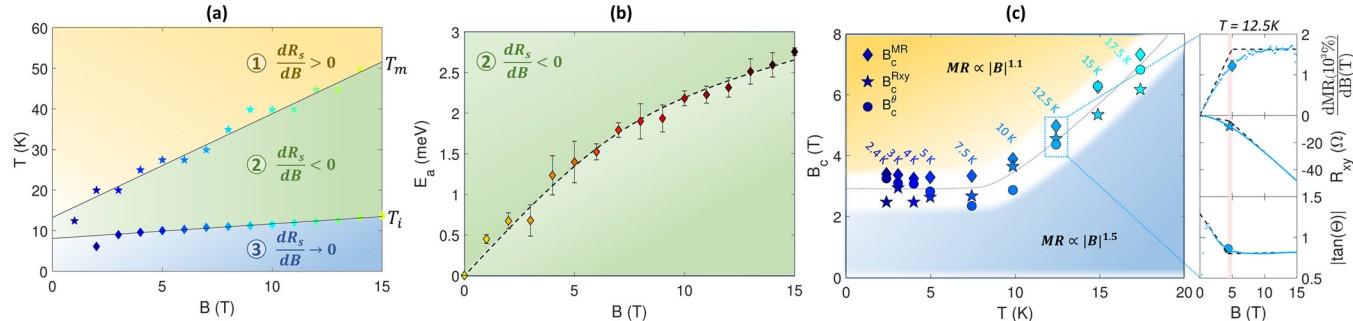

Fig. 2 | **Magnetoresistive phase diagrams. a** Magnetic field/temperature phase diagram of the three regions illustrated in Fig. 1c. **b** Thermal activation barrier ($E_a$) associated with region two extracted from an Arrhenius plot. The error bars describe the standard deviation of the linear fit in the Arrhenius plot. **c** Temperature dependence of three crossover magnetic fields ($B_c$) depicting the onset of linear magnetoresistance ($B_c^{MR}$), the inflection point of the Hall resistance ($B_c^{Rxy}$) and crossover to a field-independent tangent to the Hall angle ($B_c^{\theta}$). The insets display the field dependence of the magnetoresistive derivative (d$MR$/d$B$), the Hall resistance ($R_{xy}$) and tan($\theta$) for $T = 12.5$ K where the three crossover fields are defined by the intersection of linear fits at low and high magnetic fields.

magnetoresistance develops an approximately linear and unsaturated behavior with $MR \propto |B|^{1.08 \pm 0.01}$. Extrapolation of the high-field behavior to $B = 0$ T suggests that the $|B|^{1.5}$ and linear terms do not coexist, but rather transition from one to the other as the magnetic field is increased (Fig. S1b, c). Data for five γ-Al$_2$O$_3$/SrTiO$_3$ heterostructures showing a magnetoresistance exceeding 10,000% at 15 T are displayed in Fig. S2.

The observed magnetoresistance displays a strong temperature dependence where it gradually diminishes as the temperature is increased (Fig. 1b). It appears featureless around the cubic-to-tetragonal phase transition of SrTiO$_3$ at 105 K, but dramatically increases in magnitude below roughly 40 K, which marks the onset of quantum paraelectricity in SrTiO$_3$[45] with a concurrent large increase in the dielectric constant to above 10,000 at 2 K[46]. By examining the temperature and field dependence of the sheet resistance (Fig. 1c), the magnetoresistive behavior can be classified into three regions: When cooling the heterostructure from room to low temperature, the first region marks a metallic behavior with a monotonous decrease in the resistance for all magnetic fields (d$R_s$/d$T$>0). This behavior survives down to 2 K in absence of a magnetic field where a high residual resistance ratio of $RRR = R(295$ K$)/R(2$ K$) = 9750$ is observed, which is characteristic for high-mobility samples. As we apply an out-of-plane magnetic field of 1 T or more, we observe a resistance minimum at a characteristic temperature ($T_m$), marking the border to the second region described by a field-dependent resistance increase and d$R_s$/d$T$<0. The resistance increase grows with increasing magnetic field strengths and lowering of the temperature until an inflection point is reached. Further cooling below this inflection temperature ($T_i$) saturates the resistance in the third region. The large magnetoresistance in γ-Al$_2$O$_3$/SrTiO$_3$ forms due to the persistent zero-field metallic conductivity with a high $RRR$ providing a low value of $R_s(0$ T$)$, combined with the resistance increase in region two giving rise to a large value of $R_s(B) - R_s(0$ T$)$.

## Magnetoresistive phase diagrams

The characteristic temperatures $T_m$ and $T_i$ form a triangular field/temperature phase diagram (Fig. 2a). Here, the inflection temperature $T_i$ is extracted from the peak in d$R_s$/d$T$, which yields a value of 10 K with only a very weak magnetic field dependence. In contrast, as the magnetic field strength is increased, $T_m$ increases approximately linearly from 10 to 50 K corresponding to thermal energies between 0.9 and 4.3 meV. Very similar triangular phase diagrams were previously found to be a universal feature of XMR materials based on a comparison of semimetallic LaBi, NbSb$_2$, PtSn$_4$ and WTe$_2$ where the XMR was ascribed to charge compensation combined with mixed p-d orbital textures[12]. The striking resemblance between the phase diagrams of these materials with that of γ-Al$_2$O$_3$/SrTiO$_3$ is of particular interest considering the large chemical and electronic differences between the semimetals and the γ-Al$_2$O$_3$/SrTiO$_3$ interface. Within the temperature range of region two, ln($R_s$) scales linearly with $1/T$ suggesting a thermally activated process (Fig. S3). The linearity in the Arrhenius plot allows us to extract characteristic activation barriers taking values on the order of 1 meV with a monotonous increase with magnetic fields (Fig. 2b); this is similar to semimetallic LaSb and LaBi[12]. The physical origin of this thermally activated process will be discussed later.

A different approach to establish field/temperature phase diagrams is to explore the temperature dependence of characteristic crossover fields. The curves of d($MR$)/d$B$ generally saturate to a constant value at high magnetic fields and low temperatures, which reflects the crossover to a linear, non-saturated magnetoresistance (see top-right panel in Fig. 2c). The crossover field $B_c^{MR}$ – extracted from the intersection between linear fits to the low- and high-field regions of d($MR$)/d$B$ – is around 3 T below 10 K with an observed increase at higher temperatures (Fig. 2c). The linear fits are displayed in the top inset of Fig. 2c for $T = 12.5$ K and in Fig. S4 for all relevant temperatures. The Hall coefficient ($R_{xy}$) of the γ-Al$_2$O$_3$/SrTiO$_3$ heterostructures is linear at low and high fields, but separated by a change in slope at a crossover field ($B_c^{Rxy}$) (see middle right panel in Fig. 2c), which exhibits similar values and trends as $B_c^{MR}$ (Fig. 2c). The non-linearity was previously attributed to the anomalous Hall effect[32].

The similar field dependence of the transverse and longitudinal resistance causes a high-field saturation of the Hall angle, $\theta = \rho_{xy}/\rho_{xx} = \sigma_{xy}/\sigma_{xx}$ (lower-right panel of Fig. 2c) where $\rho_{xx}$ ($\rho_{xy}$) and $\sigma_{xx}$ ($\sigma_{xy}$) denote the longitudinal (transverse) resistivity and conductivity, respectively. By inverting the magnetoconductivity tensor, the field dependent sheet resistance may be written as:

$$R_s t = \rho_{xx} = \frac{\sigma_{xx}}{\sigma_{xx}^2 + \sigma_{xy}^2} = \frac{G}{\sigma_{xy}} \text{ with } G = \frac{\tan(\theta)}{1 + (\tan(\theta))^2} \quad (1)$$

where $t$ denotes the thickness of the electron gas. A field independent Hall angle thus entails that $G$ also becomes field independent, which gives the magnetoresistance an inverse relationship to the transverse conductivity[19]. In this case, even materials with single-band conduction displaying a linear Hall coefficient may have a linear magnetoresistance[19]. The saturation in $|\tan(\theta)|$ at a value of 0.8 occurs at the crossover field $B_c^{\theta}$ and corresponds to $|G| = 0.49$ (Fig. 2c). This value of $G$ is close to the highest value of $|G| = 0.5$. The temperature dependence of $B_c^{\theta}$ again shows a trend and magnitude similar to the other crossover fields (Fig. 2c).

## Dynamic defect tuning

An exciting aspect of forming conductivity by oxygen vacancies is the ability to dynamically tune the electronic properties by defect

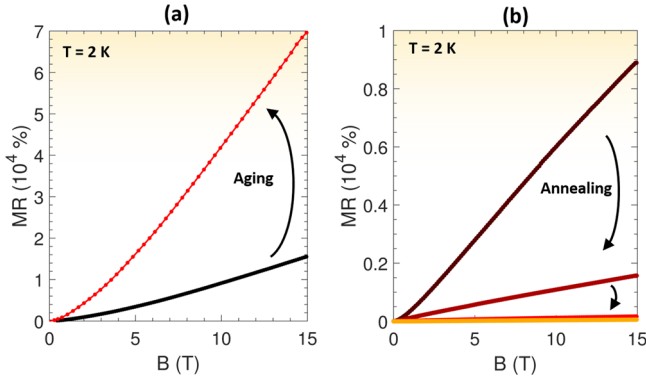

**Fig. 3 | Tuning the magnetoresistance using defect engineering.**
**a** Magnetoresistance (MR) at a temperature of 2 K for a γ-Al$_2$O$_3$/SrTiO$_3$ sample prior to and after 8 months of storage at room temperature under moderate vacuum conditions. **b** Magnetoresistance of a γ-Al$_2$O$_3$/SrTiO$_3$ sample subjected to a step-wise annealing at 200 °C in oxygen where each step is followed by cooling down the sample to 2 K to perform magnetotransport measurements.

engineering in post-synthesis treatments[34–36,47]. Prolonged storage at room temperature has previously been reported to enhance the residual resistivity ratio and electron mobility due to a diffusion of oxygen vacancies within SrTiO$_3$[34,36]. Studying the magnetoresistive properties of γ-Al$_2$O$_3$/SrTiO$_3$ under similar conditions reveals that prolonged sample storage at room temperature induces a slow aging effect. A γ-Al$_2$O$_3$/SrTiO$_3$ heterostructure with a magnetoresistance of 16,000% at 15 T and 2 K was transformed into displaying a magnetoresistance of 70,000% under the same conditions after a room temperature storage of 8 months (239 days) in a vacuum desiccator as shown in Fig. 3a.

The increase in magnetoresistance is accompanied by a decrease in the crossover field to a linear magnetoresistive regime from 4 T to 2.6 T after aging. The sample further shows a field-induced resistance upturn (region 2) both before and after aging but with the resistance upturn observed only above 4 T prior to the storage with a decrease to 2 T after storage (Fig. S5). Above this magnetic field, the activation barrier remained similar in both cases (Fig. S5). Aging the sample produced an increase in the low-temperature sheet conductance, residual resistance ratio and electron mobility (Fig. S6).

The defect dynamics can be accelerated by increasing the temperature. However, in contrast to room temperature aging for several months, annealing at 200 °C in pure oxygen gradually depletes the itinerant carriers by annihilation of oxygen vacancies, which is accompanied by monotonous drops in conductivity, residual resistivity ratio, and mobility (Figs. S7–8), as also observed elsewhere[35].

A stepwise annealing in oxygen at 200 °C for 3–12 h (see Methods) also gradually decreased the magnetoresistance by more than two orders of magnitude. The first annealing step resulted in a decrease in the magnetoresistance from 9000% to 1500% at 15 T and 2 K, followed by a reduction to 162% and 55% in the subsequent steps (Fig. 3b). Prior to annealing, the sheet resistance exhibited a weak field-induced upturn below 50 K (region two, see Fig. S7). After annealing, the sheet resistance increased across the entire temperature range and the resistance upturn was completely suppressed and replaced with featureless saturation below 20 K (Fig. S7). A drop in the value of $|\tan(\theta)|$ was also observed (Fig. S8).

## Magnetoresistive scaling

The post-synthesis variations induced by room temperature aging and annealing in oxygen are compared in Fig. 4b, c with a growth-induced approach where the oxygen vacancies are engineered by varying the oxygen partial pressure during growth. The as-deposited zero-field mobility ($\mu_0$) and *MR* show a dome-shaped dependence of the sheet carrier density where a maximum $\mu_0 > 100,000$ cm$^2$/Vs ($RRR = 10^4$) and

$MR = 80,000\%$ are obtained at $n_s = 5 \cdot 10^{14}$ cm$^{-2}$. Sheet carrier densities one order of magnitude higher than this value result in bulk conductivity from growth-induced oxygen vacancies[48]. We further observe a concurrent increase in both the mobility and magnetoresistance after room temperature aging, but with the sheet carrier density being largely unaffected. In contrast, both the mobility, magnetoresistance and carrier density drop monotonously as annealing in oxygen is performed.

Next, we utilized scanning superconducting quantum interference devices (SQUIDs) to image the local current contributions in γ-Al$_2$O$_3$/SrTiO$_3$ by imaging the magnetic stray field formed when driving a current through the heterostructure, as also done elsewhere[48]. In Fig. 4a, we present scanning SQUID measurements performed in absence of an applied magnetic field on four γ-Al$_2$O$_3$/SrTiO$_3$ heterostructures with carrier densities varying by a factor of 30. See also large area scans over 0.5 mm wide regions in Fig. S9. The inhomogeneous magnetic stray fields detected by the scanning SQUID directly reflect an inhomogeneous current flow in the sample. For low sheet carrier densities ($n_s \leq 10^{14}$ cm$^{-2}$), stripes from ferroelastic domain walls in SrTiO$_3$[48,49] as well as hole-like patterns were observed in the current flow. At higher carrier densities ($n_s \geq 5 \cdot 10^{14}$ cm$^{-2}$), these sharp modulations vanish and instead, slowly varying modulations in the magnetic stray field appear with a length scale of a few tens of micrometer. The disappearance of measurable stripes was attributed to either an extended depth distribution of the conducting electrons or an enhanced electronic screening in the high-density system[48]. We note that the length scale of the stray field variations is broadened by the distance between the SQUID pick-up-loop and the current as well as the point-spread-function of the 1.8 μm pick-up loop, and hence submicron features may not be resolved. Nonetheless, it is striking that the highest magnetoresistance and mobility are observed at the onset of the high-carrier density regime where the local current flow has transitioned into causing a weakly modulating stray field rather than stripe-like features.

To shed further light on the positive correlation between the carrier density and magnetoresistance, we revisit our previous resonant angle-resolved X-ray photoemission spectroscopy (ARPES) measurements probing the momentum-resolved band structure of the buried interface[31]. Figure 4d, e replots this data in the new context of exploring the role of the band structure in determining the magnetoresistive behavior. Consistent with other SrTiO$_3$-based material systems[50–54], the γ-Al$_2$O$_3$/SrTiO$_3$ heterointerfaces are characterized by a large band gap and a Fermi level located within a conduction band composed of heavy *n*-type $t_{2g}$ bands. No features associated with *p*-type conduction were observed. The γ-Al$_2$O$_3$/SrTiO$_3$ heterointerfaces feature a band reordering where anisotropic heavy $d_{xz}$ and $d_{yz}$ bands are lowest in energy[31,55,56] in sharp contrast to typical SrTiO$_3$-based systems where the $d_{xy}$ band is the first available conduction band[50–54]. The heavy bands are inferred to have a larger spatial extent[31,54,57,58]. The effective masses and Fermi velocities are 0.4 $m_e$ & 9.8 $m_e$ and 0.069 Å$^{-1}$ & 0.34 Å$^{-1}$ in the light and heavy direction, respectively, with a strong polaronic mass enhancement. Here, $m_e$ denotes the free electron mass. Similar band structures were acquired on samples with sheet carrier densities varying by a factor of 20, bridging the regimes of low and high magnetoresistance (Fig. 4c-e). The insensitivity to large sheet carrier density variations is likely caused by adding carriers deeper in SrTiO$_3$, beyond the detection limit of ARPES. In contrast to conventional gapless XMR materials with light bands, charge compensation and small effective masses[4], the γ-Al$_2$O$_3$/SrTiO$_3$ heterostructures thus features heavy bands and solely *n*-type conductivity in a large band-gap structure.

We investigate the similar trends of the mobility and magnetoresistance in Fig. 4b, c using the temperature dependence on a single sample where lowering the temperature both boosts the zero-field mobility and magnetoresistance. The magnetoresistance at 15 T scales

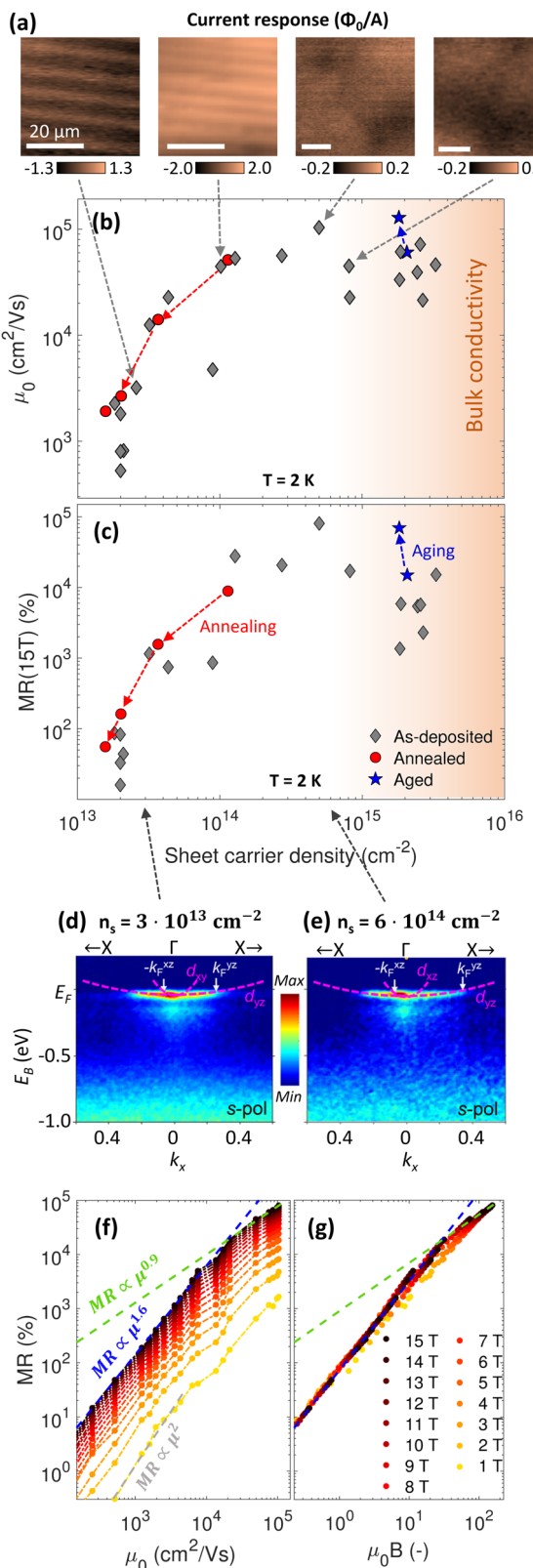

**Fig. 4 | Magnetoresistive scaling. a** Magnetic flux measured from an alternating current applied to the four $\gamma$-$Al_2O_3$/$SrTiO_3$ heterostructures with varying carrier densities. The magnetic flux is measured with a scanning superconducting quantum interference device with a pick-up loop of 1.8 μm. The scale bar is 20 μm in all cases. **b** Comparison of the zero-field mobility ($\mu_O$) and (**c**) magnetoresistance at 15 T as a function of the sheet carrier density for a variety of $\gamma$-$Al_2O_3$/$SrTiO_3$ samples where the oxygen growth pressure and post-treatment are varied. The figure compiles mobility data also presented in ref. [48] with present data, but all samples were grown in the same pulsed laser deposition chamber by the same person. **d**, **e** Magnetoresistance at magnetic fields ranging from 1–15 T displayed as a function of the zero-field mobility and scaled as $\mu_O B$. Figures (**d**, **e**) were adapted with permission from ref. [31], respectively.

for the high-temperature case of $B = 1$ T in Fig. 4f. The scaling behaviors of the two low-temperature regimes ($MR \propto \mu_0^{1.6}$ and $MR \propto \mu_0^{0.9}$) are similar to the scaling behavior in the phase diagram presented in Fig. 2c ($MR \propto B^{1.5}$ and $MR \propto B^{1.1}$). This motivates scaling the magnetoresistance as a function of $\mu_0 B$, which is observed to nearly collapse all the curves on top of each other with the linear onset occurring at $\mu_0 B > 25$ (Fig. 4g). As the carrier density is fairly temperature independent in $\gamma$-$Al_2O_3$/$SrTiO_3$, this scaling resembles Kohler's rule stipulating that the magnetoresistance scales as $B/\rho_0$ (see Kohler plots in Fig. S11).

## Origin of the extreme magnetoresistance

A detailed description of the origin of the unsaturated linear magnetoresistance and its tunability is provided in Supplementary Section 10 with a brief outline provided here. Despite the similar triangular phase diagram, the origin of XMR in $\gamma$-$Al_2O_3$/$SrTiO_3$ contrasts that of archetypical high-mobility XMR materials[4]. Instead, the linearity of the magnetoresistance with an onset of large $\mu_0 B$ values and the saturating Hall angle can be described by a semiclassical guiding center transport model[19]. The prerequisites for the guiding center model proposed by Song et al.[19] are all fulfilled in $\gamma$-$Al_2O_3$/$SrTiO_3$, including (1) high mobility, (2) 3D or quasi−2D transport justified by the high sheet carrier density and the ARPES and SQUID data, (3) weak disorder assessed by the Fermi energy (tens of meV) exceeding the thermal energy associated with relaxing the high-resistive state (a few meV), and (4) slowly varying disorder potential justified by the SQUID measurements and the high dielectric constant of $SrTiO_3$ (see Supplementary Section 10 for further details).

The linear magnetoresistance is mathematically a consequence of the field-saturating $|\tan(\theta)|$ as described by Eq. 1. A physical guiding center picture is provided in Fig. 5. For itinerant charge carriers strictly confined to a 2D sheet with a slowly varying disorder potential landscape, semiclassical transport causes the carriers to undergo rapid cyclotron motion while following an overall guiding center motion with closed orbits along the equipotential contours of the disorder potential (Fig. 5a). The carriers can escape the closed orbits by an out-of-plane ($z$) movement (Fig. 5b). If the disorder potential energy in some places exceeds the kinetic energy of the electrons, out-of-plane movement is restricted and the current gets squeezed along $z$. This temporary in-plane constraint is associated with a slow guiding center velocity and a resulting linear magnetoresistance[19]. Providing thermal energy to the system enables free movement across layers (Fig. 5c), which eliminates the linear, high-resistive state, consistent with the thermal relaxation in Fig. 2b. The temperature increase also enhances scattering, which may cause a departure from the semiclassical transport regime. In this view, the room temperature aging promotes the linear XMR by reducing scattering and widening the electron depth distribution as oxygen vacancies distribute deeper into $SrTiO_3$ (Fig. 5b). A detrimental effect is observed after annealing at 200 °C as the effective disorder is increased, which is manifested in a $\tan(\theta)$ decrease (Fig. S8d) as predicted for 3D Dirac materials[19]. The intimate link between defect engineering and the XMR is further described from a mechanistic perspective in Supplementary Section 10.

as $MR(15\,T) \propto \mu_0^{1.64 \pm 0.04}$ below $\mu_0 < 20{,}000$ cm$^2$/Vs (Fig. 4f), which is reasonably consistent with how the magnetoresistance varies with mobility across multiple samples (Fig. S10). At higher mobilities, the scaling behavior transitions into a regime characterized by $MR(15\,T) \propto \mu_0^{0.88 \pm 0.04}$. Decreasing the magnetic field shifts the crossover between the two regimes to higher zero-field mobilities. Eventually, a quadratic magnetoresistance behavior emerges as depicted

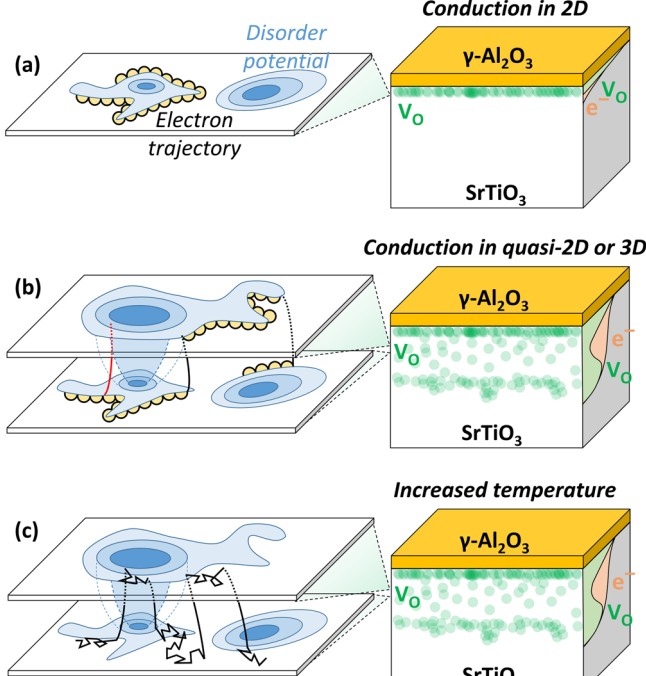

**Fig. 5 | Schematics of the proposed mechanism.** Schematic illustrations of the semiclassical magnetotransport in high perpendicular magnetic fields when high-mobility electrons (**a**) are strictly confined in 2D, (**b**) are restricted to move vertically by a disorder potential in some locations (red vertical line) but allowed in other locations (black vertical lines) and (**c**) are heated to reduce the mean-free path with the thermal energy exceeding the disorder potential variations, which leads to unrestricted vertical movement.

The chemical, structural, and functional dissimilarity of the oxide platform compared to conventional XMR materials thus brings new insight into the origins of XMR and paves a new way for its tunability. The moderate diffusion barrier for oxygen vacancy donors ($\sim 0.5$ eV[35]) combined with an inhomogeneous oxygen vacancy depth distribution resulting from the broken symmetry at the heterointerface enable the tunability of both the electron mobility and disorder potential through dynamic defect engineering. Lateral and vertical spatial modifications of the oxygen vacancy locations using electromigration[33,59,60] and selective area epitaxy[61,62] are exciting perspectives for on-demand tailoring of the disorder landscape and the associated XMR. We deduce the length scale of the disorder to be at least hundreds of nanometers (see Supplementary Section 10), making it feasible to both artificially tailor the disorder landscape and use scanning magnetometry to visualize the resulting field-dependent electron trajectory[63]. Overall, the advances made here demonstrate the prospects of using oxide heterostructures for studying and tailoring XMR.

## Methods
### Heterostructure synthesis
The γ-Al₂O₃/SrTiO₃ samples were prepared using pulsed laser deposition of γ-Al₂O₃ on TiO₂-terminated substrates[60]. The SrTiO₃ substrates were prepared by immersion in water (20 min at 70 °C) and H₂O:HCl:HNO₃ = 16:3:1 (20 min at 70 °C) followed by annealing in pure oxygen for 1 h at 1000 °C. The γ-Al₂O₃ films with a thickness of 2.8 nm were deposited under optimized growth conditions[48] at 650 °C in a low oxygen partial pressure of $10^{-5}$ mbar. The pulsed laser deposition used a 248 nm KrF laser operating at 1 Hz with a laser fluency of 3.5 J/cm². After deposition, the samples were cooled to room temperature at the deposition pressure without a post-annealing step. The samples in Fig. 4b-c followed the same deposition protocol, but with variations in

the oxygen partial pressure in the range of $10^{-4} - 10^{-6}$ mbar to tune the sheet carrier density.

### Transport measurements
The conducting heterointerfaces were contacted electrically using wedge wire bonding with aluminum wires. DC magnetotransport measurements were conducted in van der Pauw geometry in magnetic fields up to 15 T and temperatures ranging from room temperature to 2 K. Only for the case presented in Fig. S14 was magnetotransport measured in a 4-terminal configuration with electrical contacts arranged equidistantly on a straight line in the middle of the sample. To avoid intermixing of the magnetoresistance and Hall curves, the measurements were symmetrized and anti-symmetrized for the magnetoresistance and Hall data, respectively. The main features of the transport measurements remained unchanged during this symmetry operation. A small anisotropy of around ± 10% was observed below 7 K when measuring the magnetoresistance along the horizontal and vertical configuration, respectively (Fig. S16). The carrier density and zero-field mobility ($\mu_0$) are extracted from the linear part of the Hall coefficient at weak fields where the magnetoresistance is weak (see ref. 32 for further discussions). To investigate the effect of aging, we performed temperature-dependent magnetotransport on the γ-Al₂O₃/SrTiO₃ heterostructures in the van der Pauw geometry prior and after prolonged storage of 239 days in a vacuum desiccator providing a clean environment with a slight vacuum on the order of 0.5 bar. The effect of annealing in oxygen was also investigated by placing the γ-Al₂O₃/SrTiO₃ heterostructure in pure oxygen at a temperature of 200 °C. Temperature-dependent magnetoresistance measurements were performed in van der Pauw geometry at subsequent annealing steps with the annealing times at each step described in Fig. S7.

### Scanning SQUID measurements
The scanning SQUID measurements were conducted using a planar gradiometer SQUID to detect the magnetic stray field at 4 K in a 1.8 μm pick-up loop connected electrically to the SQUID[48]. The pick-up loop was scanned over the atomically flat sample surface to form a 2D map of the stray field from an alternative current in the γ-Al₂O₃/SrTiO₃ sample. The detected magnetic flux is given by $\Phi = \int g(x,y)\bar{B} \cdot d\bar{a}$ where $g(x,y)$ and $d\bar{a}$ denote the point-spread-function of the pick-up loop and an infinitesimal areal vector pointing normal to the plane of the pick-up loop. More information on the technique can be found elsewhere[32,64,65]. The scanning SQUID measurements were conducted on the same four samples as in ref. 48, but on different areas of the samples and, in the case of Fig. S9, on wide areas spanning ~0.5 mm.

### ARPES measurements
The ARPES measurements were adapted from ref. 31 where the measurements were conducted at 12 K at the ADRESS beamline at the Swiss Light Source, Paul Scherrer Institute using soft X-ray to resonantly excite the Ti L-edge. The data were acquired using $10^{13}$ photons/s with a spot size of 30·75 μm², an energy resolution of around 40 meV and an angular resolution of 0.1°. The intensity of the signal increased during X-ray irradiation, but similar to previous reports on SrTiO₃-based materials systems[31,66,67], the band dispersion and band population remained constant. Additional ARPES measurements on the γ-Al₂O₃/SrTiO₃ heterostructures are found in ref. 31.

## Data availability
The data generated in this study have been deposited in the data.d-tu.dk database accessible at https://doi.org/10.11583/DTU.25611375[68].

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

## Acknowledgements
D.V.C. and T.S.S. acknowledge the support of Novo Nordisk Foundation NERD Programme: New Exploratory Research and Discovery, Superior Grant NNF21OC0068015. D.V.C., T.D.P. and N.P. acknowledge the support of Novo Nordisk Foundation Challenge Programme 2021: Smart nanomaterials for applications in life-science, BIOMAG Grant NNF21OC0066526. N.P. acknowledges the support from the ERC Advanced "NEXUS" Grant 101054572. B.K. acknowledges support from ERC COG no. 866236, ISF- 228/22, DIP KA 3970/1-1, and COST action SUPERQUMAP CA 21144. The authors acknowledge Alla Chikina for her input to the acquisition and analysis of the ARPES data in relation to ref. 31, Yiftach Frenkel for his help on the acquisition of scanning SQUID images, and Shai Rabkin for data analysis.

## Author contributions
D.V.C. designed the experiments. D.V.C. prepared the samples. D.V.C. performed the magnetotransport, aging and annealing experiments. D.V.C., V.S. and N.P. performed the ARPES measurements. D.V.C. and B.K. performed the SQUID measurements. D.V.C. performed the data analysis, except for the ARPES data conducted by V.S. and Alla Chikina. D.V.C., T.S.S. and T.D. Pomar contextualized the obtained results. D.V.C. wrote the manuscript with input from all authors. All authors discussed the results and commented on the manuscript.

## Competing interests
The authors declare no competing interests.
