## [Peer Review File · Nature Communications]

Reviewers' Comments:

Reviewer #2:

Remarks to the Author:

The study explores the emergence of a linear extreme magnetoresistance (XMR) at the interface between γ -Al₂O₃ and SrTiO₃. Based on several experimental characterizations, the authors conclude that the XMR in γ -Al₂O₃/SrTiO₃ is not strongly linked to the band structure, but arises from weak disorder, shedding light on a novel perspective on XMR mechanisms and tunability. I find the novelty of the results may not be suitable for nature communications, especially for the ARPES data which has been presented in previous publications. I can not recommend the publication of this manuscript in Nature Communications.

Reviewer #3:

Remarks to the Author:

In this paper, Christensen et al. presents a thorough and rigorous study showing one of the largest extreme magnetoresistance (XMR) in a particular gamma-Al₂O₃/SrTiO₃ heterostructure instead of normal semimetals, investigating magnetotransport, microscopic current imaging, Angle-Resolved X-ray Photoemission Spectroscopy (APRES), vacancy-tuning, etc., and providing a detailed description of the origin of this XMR. The finding in the oxide system is interesting, the data and plot are of high-quality and multidisciplinary, and the explanation of the mechanism looks convincing. I will recommend to publish in Nature Communications, but before publication I just have some minor comments on this work:

(1) In my opinion, some figures could be replaced in the supplementary information (SI), like Fig.2(b) and Fig.S3(b), Fig.2(c) and Fig.S4(d). If the authors could either replace or edit the figures in SI to provide more information than the main text, that would be nice.

For example, Fig.S3(a) shows the application of Arrhenius Law and use blue dashed lines to present the region you obtain the slope in $\ln(R_s)$, panel (b) is clear but repeated in the main text, maybe you could change panel (b) to show all the extended dashed line and to present the tendency how the slope changes with field, or add the equation in SI for the relation between E_a and slope/add a table how you extract each slope & E_a .

(2) Some plots could be improved: Fig.1(a) labels for temperatures, Fig.S4(c) some of the circle markers are not very clear, also the plot looks a bit messy (the idea is clear though)

(3) If there's a method to quantify the oxygen vacancies, or compare the difference in vacancies between different samples before/after aging, annealing, etc. (SEM for example), this could provide direct evidence for the connection between vacancies and its magnetotransport properties in tunability.

To summarize, the authors have presented XMR, a conventional magnetotransport behavior in a pretty unconventional system, with very different chemical, structural properties, also made many attempts and presented nice work for its tunability and successfully combined this with the mechanism. This fairly new system opens up possibilities for exploring XMR and its applications. Therefore, I recommend this manuscript for publication in Nature Communications.

The following comprise our point-to-point response to the comments from the two referees:

Reviewer #2 (Remarks to the Author):

Reviewer comment:

The study explores the emergence of a linear extreme magnetoresistance (XMR) at the interface between γ -Al₂O₃ and SrTiO₃. Based on several experimental characterizations, the authors conclude that the XMR in γ -Al₂O₃/SrTiO₃ is not strongly linked to the band structure, but arises from weak disorder, shedding light on a novel perspective on XMR mechanisms and tunability. I find the novelty of the results may not be suitable for nature communications, especially for the ARPES data which has been presented in previous publications. I can not recommend the publication of this manuscript in Nature Communications.

Reply:

We thank the reviewer for the comments. We agree with the reviewer that the band structures measured by ARPES are not by themselves novel; nonetheless, they bring new insight into the mechanism of XMR, which has not been considered for oxide materials before. We would also like to note that the ARPES data only constitute a small role (~5%) of the manuscript, which serves to exclude the charge compensation mechanism conventionally used to explain the origin of XMR. The main novelties of the manuscript relate to the following key findings:

1. We discover a large XMR of ~10⁵% at 15 T in the strongly correlated γ -Al₂O₃/SrTiO₃ heterostructure with a mobility exceeding 100,000 cm²/Vs. This is not only the first report of XMR at a heterointerface, but the chemical dissimilarity with conventional XMR materials also brings new insights into the origin of XMR.
2. We show for the first time that XMR can be tuned using (dynamic) defect engineering, and present three pathways for realizing this defect engineering concept. Here, we show that this can result in a dynamic variation of the XMR by several orders of magnitude.
3. We combine detailed magnetotransport analysis, dynamic defect engineering, direct visualization of current distributions using quantum magnetic microscopy, and momentum-resolved band structure measurements to understand the origin of the XMR. This is the first approach combining these techniques to get a comprehensive view of the XMR origin (see Supplementary Section 10 for details).
4. We assign the origin of XMR to a semiclassical guiding center motion of electrons in an inhomogeneous conductor, and build on this model to mechanistically explain how defect engineering can tune the XMR.

In the updated manuscript and supplementary materials, we now also provide new scanning SQUID images of γ -Al₂O₃/SrTiO₃ (Fig. 4a) including large-area (~0.5 mm) scanning SQUID images (Fig. S9 in the supplementary materials), which illustrate how the current distributes at the micrometer scale across wide areas in oxide heterostructures. These scanning SQUID measurements are performed on samples with both low and high magnetoresistance and support the XMR mechanism proposed in the manuscript.

We have emphasized the above-mentioned points of novelty in the revised manuscript and hope that the reviewer will reconsider his/her view - thank you for your consideration.

Reviewer #3 (Remarks to the Author):

Reviewer comment:

In this paper, Christensen et al. presents a thorough and rigorous study showing one of the largest extreme magnetoresistance (XMR) in a particular gamma-Al₂O₃/SrTiO₃ heterostructure instead of normal semimetals, investigating magnetotransport, microscopic current imaging, Angle-Resolved X-ray Photoemission Spectroscopy (APRES), vacancy-tuning, etc., and providing a detailed description of the origin of this XMR. The finding in the oxide system is interesting, the data and plot are of high-quality and multidisciplinary, and the explanation of the mechanism looks convincing. I will recommend to publish in Nature Communications, but before publication I just have some minor comments on this work:

Reply:

We thank the reviewer for the in-depth review of the main text and supplementary material. We address your comments below.

Reviewer comment:

(1) In my opinion, some figures could be replaced in the supplementary information (SI), like Fig.2(b) and Fig.S3(b), Fig.2(c) and Fig.S4(d). If the authors could either replace or edit the figures in SI to provide more information than the main text, that would be nice.

For example, Fig.S3(a) shows the application of Arrhenius Law and use blue dashed lines to present the region you obtain the slope in $\ln(Rs)$, panel (b) is clear but repeated in the main text, maybe you could change panel (b) to show all the extended dashed line and to present the tendency how the slope changes with field, or add the equation in SI for the relation between E_a and slope/add a table how you extract each slope & E_a .

Reply:

We thank the reviewer for the comment and agree that Fig. S3b and Fig. S4d in the supplementary material are repetitive. Following the suggestion from the reviewer, we have modified Figs. S3a-b to include the Arrhenius equation, which is overlaying a visual guide on transforming a selected linear line segment into the corresponding thermal activation barrier. As the slope of the linear Arrhenius plot is proportional to the thermal activation barrier, Fig. S3b captures both the field dependence of the slope and the activation barrier. The figure caption has also been changed accordingly.

Regarding Figure S4, we have reformatted this figure, so it now more clearly shows the flow from Figure S4a-c to Figure S4d. In addition, we have also deleted the inset of Figure S4d, which is identical to that of Figure 2c in the main text.

Reviewer comment:

(2) Some plots could be improved: Fig.1(a) labels for temperatures, Fig.S4(c) some of the circle markers are not very clear, also the plot looks a bit messy (the idea is clear though)

Reply:

The temperature labels in Figure 1a have been fixed. In addition, we have improved the clarity of the markers in Figures S4b-c and provided a zoom-in of the circle markers in Figure S4c.

Reviewer comment:

(3) If there's a method to quantify the oxygen vacancies, or compare the difference in vacancies between different samples before/after aging, annealing, etc. (SEM for example), this could provide direct evidence for the connection between vacancies and its magnetotransport properties in tunability.

Reply:

Thank you for the great question. Quantifying oxygen vacancy defects non-invasively in oxide heterostructures is a challenging task owing to their buried nature, low concentration, and dynamic behavior. Methods such as SEM unfortunately are not capable of quantifying oxygen vacancies; see e.g. Appl. Phys. Lett. 116, 120505 (2020) where we discuss available tools for detecting oxygen vacancies.

Tools for monitoring low concentrations of oxygen vacancies often rely on the well-established notion that the creation of oxygen vacancies in oxides results in two additional electrons per oxygen vacancy. This renders SrTiO_3 conducting, and, therefore, the electrical conductivity (carrier density) is routinely used to investigate the presence and amount of oxygen vacancies in SrTiO_3 -based heterostructures, as transport characterization is a tool that is non-invasive and sensitive to minute oxygen vacancy concentrations.

We have also established direct defect-property relationships in the $\gamma\text{-Al}_2\text{O}_3/\text{SrTiO}_3$ heterostructures where 1) the oxygen vacancies are responsible for all conducting carriers in the heterostructures, and 2) the scattering landscape formed by the oxygen vacancies determines the low-temperature electron mobility. The former is established using several techniques, including in-situ transport measurements in oxygen or oxygen-free atmospheres during the $\gamma\text{-Al}_2\text{O}_3$ growth^{1,2}, post-growth annealing at various temperatures and environments^{3,4}, high-temperature equilibrium conduction measurements⁵, and angle-resolved photoemissions spectroscopy^{6,7}. The latter is established through annealing-dependent magnetotransport⁸, scanning SQUID measurements^{8,9}, angle-resolved photoemissions

spectroscopy⁶, and numerical modelling^{6,10}. Hence, we can directly use the electron density as a measure of the oxygen vacancy defect density, and the electron mobility as a measure of the defect positions.

Next, we can consider the three ways for tuning the magnetoresistance, namely using different growth conditions, post-growth annealing in oxygen, and sample aging (Figure 4b-c). Increasing the oxygen partial pressure during growth as well as performing post-annealing in an oxygen-containing environment are both well-known to reduce the amount of oxygen vacancies^{3-5,7}. This is consistent with the dramatical reduction in the electron density, which, in turn, degrades the magnetoresistance by several orders of magnitude. In contrast, sample aging increases the magnetoresistance while resulting in only small changes in the carrier density, hence testifying that the oxygen vacancy defect concentration remains roughly unchanged but the oxygen vacancies relocate^{7,10}. In addition, we have further correlated the change in electronic transport properties with the thermal activation barrier during annealing experiments³, and found that the activation barrier matches that of oxygen vacancy movement in SrTiO₃.

In conclusion, there is a very strong link between the oxygen vacancies and the tunable magnetoresistance. We have provided a detailed explanation of the three defect engineering pathways (growth variation, annealing & aging) and their direct impact on magnetotransport in Supplementary Section 10 and referred to this section in the main text.

Reviewer comment:

To summarize, the authors have presented XMR, a conventional magnetotransport behavior in a pretty unconventional system, with very different chemical, structural properties, also made many attempts and presented nice work for its tunability and successfully combined this with the mechanism. This fairly new system opens up possibilities for exploring XMR and its applications. Therefore, I recommend this manuscript for publication in Nature Communications.

Reply:

We thank the reviewer for the detailed assessment and for appreciating our manuscript.

References:

1. von Soosten, M. *et al.* On the emergence of conductivity at SrTiO₃-based oxide interfaces – an in-situ study. *Sci Rep* **9**, 18005 (2019).
2. Hvid-Olsen, T. *et al.* Spatial control of the conductivity in SrTiO₃-based heterointerfaces using inkjet printing. *J. Phys. Energy* **4**, 044005 (2022).
3. Christensen, D. V. *et al.* Controlling the carrier density of SrTiO₃-based heterostructures with annealing. *Advanced Electronic Materials* **3**, 1700026 (2017).
4. Steegemans, T. *et al.* Tuning Oxide Properties by Oxygen Vacancy Control During Growth and Annealing. *JoVE* 58737 (2023) doi:10.3791/58737.
5. Gunkel, F. *et al.* Thermodynamic Ground States of Complex Oxide Heterointerfaces. *ACS Applied Materials & Interfaces* **9**, 1086–1092 (2017).
6. Chikina, A. *et al.* Band-Order Anomaly at the γ -Al₂O₃/SrTiO₃ Interface Drives the Electron-Mobility Boost. *ACS Nano* **15**, 4347–4356 (2021).
7. Schütz, P. *et al.* Microscopic origin of the mobility enhancement at a spinel/perovskite oxide heterointerface revealed by photoemission spectroscopy. *Physical Review B* **96**, 161409 (2017).
8. Christensen, D. V. *et al.* Electron Mobility in γ -Al₂O₃/SrTiO₃. *Physical Review Applied* **9**, 054004 (2018).
9. Bjørlig, A. V., Christensen, D. V., Erlandsen, R., Pryds, N. & Kalisky, B. Current Mapping of Amorphous LaAlO₃/SrTiO₃ near the Metal–Insulator Transition. *ACS Appl. Electron. Mater.* **4**, 3421–3427 (2022).
10. Zurhelle, A. F., Christensen, D. V., Menzel, S. & Gunkel, F. Dynamics of the spatial separation of electrons and mobile oxygen vacancies in oxide heterostructures. *Phys. Rev. Materials* **4**, 104604 (2020).

Reviewers' Comments:

Reviewer #3:

Remarks to the Author:

I would like to thank the authors for their thorough and thoughtful response to both my comments and those of the other reviewer. In their response, they underscore the main idea and novelty of their work: providing new insights into the origin of XMR in a rather unconventional system, and demonstrating the tunability of XMR through dynamic defect engineering. Through their revisions to the manuscript and supplementary materials, the authors have significantly clarified the motivation behind their work.

I am inclined to recommend the publication of this work in Nature Communications. However, given the contrasting opinion of the other referee regarding its novelty, I would suggest seeking a third reviewer's opinion if the stance of the other reviewer remains unchanged.